# OpenReview forum: "MemOrb: A Plug-and-Play Verbal-Reinforcement Memory Layer for E-Commerce Customer Service"
_ICLR.cc/2026/Conference — ICLR 2026 Conference Withdrawn Submission_

### Official Review · Reviewer_V3Uj · 2025-10-15

**Soundness:** 1
**Presentation:** 1
**Contribution:** 1
**Rating:** 0
**Confidence:** 3

**Summary:**

The paper proposes a system for "policy-reflection distillation" for customer service. This system is based on emotion tagging and policy reflection modules.

**Strengths:**

The paper correctly identifies some areas where existing LLM-based agents fail.

**Weaknesses:**

The paper is not yet ready for detailed review. The main issues are as follows:
-- Major writing problems that limit the clarity of the paper. See e.g. Figure 1 and its caption
-- The paper contains almost no technical detail about the approach, beyond very limited pseudocode. This makes it mostly impossible to assess the technical relevance or methodological contribution of the paper
-- The topic, while ML-related, is largely not relevant to ICLR
-- The baselines are far too limited for the method to be properly evaluated
-- Bibliography is nearly entirely just arxiv links, I assume automatically generated

**Questions:**

It would be great if the authors try to clarify the main technical contribution of the paper and its relevance to ICLR. I realize this is a lot to ask during a rebuttal phase but the paper does not seem quite ready to review without more technical detail.

---

### Official Review · Reviewer_7two · 2025-10-27

**Soundness:** 2
**Presentation:** 2
**Contribution:** 2
**Rating:** 2
**Confidence:** 4

**Summary:**

The paper introduces MemOrb, a method designed to enhance LLM agents through policy-level reflections in multi-trial scenarios. Additionally, the authors expand the ECom-Bench dataset by adding 77 new clothing-domain tasks, resulting in a total of 130 realistic multi-turn customer service tasks.

**Strengths:**

1. The idea of distilling multi-turn interactions into compact strategic representations is interesting and potentially useful for improving agent performance.

2. The paper provides detailed experimental analysis of success rate trends across different trial counts (Figure 3).

**Weaknesses:**

1. Experiments are conducted only on ECom-Bench, which includes 144 tasks. It remains unclear whether MemOrb generalizes effectively to other LLM agents benchmarks.
2. Compared with Reflexion [1], the methodological novelty of MemOrb appears limited or insufficiently highlighted.
3. The paper lacks detail on how the additional 77 clothing-domain tasks were constructed, including data sources, task diversity, or annotation quality.
4. As shown in Table 2, Doubao-Seed-1.6-Thinking-MemOrb surpasses Doubao-Seed-1.6-Thinking only when the number of trials exceeds 5. However, it underperforms for T1–T4, suggesting that the method is less effective in low-trial or single-pass scenarios, which are also common in practical settings.

[1] Reflexion: Language Agents with Verbal Reinforcement Learning

**Questions:**

1. The paper claims that MemOrb is motivated by tasks requiring stability and consistency. However, it is not entirely clear why multi-trial settings are important in the e-commerce domain. In customer service scenarios, users typically expect the agent to succeed in a single interaction, making the one-pass success rate more relevant than multi-trial performance.

---

### Official Review · Reviewer_fxtT · 2025-10-30

**Soundness:** 1
**Presentation:** 3
**Contribution:** 1
**Rating:** 0
**Confidence:** 5

**Summary:**

The paper “MemOrb: A Plug-and-Play Verbal-Reinforcement Memory Layer for E-Commerce Customer Service” (under review at ICLR 2026) proposes a lightweight, schema-free memory system that allows frozen LLM-based agents (i.e., without any fine-tuning) to continuously improve through reflection across user sessions.

**Strengths:**

Self-Reflection is an interesting idea and worth exploring.

**Weaknesses:**

1. **Inaccurate summarizations and comparisons:** In Table 1, MemGPT is not simply using key-value pairs and not just raw dialogues. They have core memory which is the summarization and extracted infomation, they also use `SQLite` or `PostGreSQL` as the storage, while the paper only says "Key-Value" in the column Storage. Also in MemGPT they do have the ability to rewrite the memory. In their codebase there is a function called `core_memory_replace` and `core_memory_rewrite`. If your "ReWrite" is talking about rewriting the query, then the table is even more incorrect. MemGPT has agentic search which definitely has the ability to rewrite the query.

2. **Limited Novelty:** Basically the paper is proposing that we can (1) save the raw trajectories; (2) do some self-reflection after each trajectory; (3) retrieve relevant trajectories. The novelty and design remains trivial and sounds like a simple pipeline every company would easily think of during the applications. In the experiments, there are only 130 tasks in total which makes the evaluation results highly unstable and unconvincing.

3. **Limited Evaluation Datasets:** If the method is simple I would expect this method to have much more powerful performance across various tasks such as Long-Horizon Agent Tasks like TAC[1], Mind2Web[2], SWE-Bench-Pro[3], etc, instead of only showing the performance on a not-so-popular Ecom-bench. This is a research paper, not a technical report in the Industry.

4. **Limited Baselines:** This paper compared with **zero** memory-related baselines. Even though they mentioned in the introduction about the limitations of existing memory-augmented methods, the limitations of long-context methods, they compared with none of them in the experiments.

5. **Missing Citations**: Many recent papers about agent memory systems are not cited [4,5,6,7,8,9] (there should be much more than these such as Agent Workflow Memory, Mem-p, ...). Also in Line 152, this paper mentioned that "MemOrb offers a lightweight, plug-and-play solution that improves the performance of LLM-based agents without the need for frequent model updates or **large-scale retraining,**" However they did not mention any related works about the memory methods that require "large-scale retraining" (some representative works are MemoryLLM[10], M+[11]). They can either not mention this in the introduction, or they have to cite related papers to justify the statement.

[1] TheAgentCompany: Benchmarking LLM Agents on Consequential Real World Tasks.
[2] Mind2Web: Towards a Generalist Agent for the Web.
[3] SWE-Bench Pro: Can AI Agents Solve Long-Horizon Software Engineering Tasks?
[4] Nemori: Self-Organizing Agent Memory Inspired by Cognitive Science.
[5] MIRIX: Multi-Agent Memory System for LLM-Based Agents.
[6] EgoMem: Lifelong Memory Agent for Full-duplex Omnimodal Models.
[7] Zep: A Temporal Knowledge Graph Architecture for Agent Memory.
[8] MemoRAG: Boosting Long Context Processing with Global Memory-Enhanced Retrieval Augmentation.
[9] HippoRAG: Neurobiologically Inspired Long-Term Memory for Large Language Models.
[10] MEMORYLLM: Towards Self-Updatable Large Language Models.
[11] M+: Extending MemoryLLM with Scalable Long-Term Memory.

**Questions:**

I don't have any questions.

**Details Of Ethics Concerns:**

I do not have any ethical concerns.

---

### Official Review · Reviewer_3tip · 2025-10-30

**Soundness:** 2
**Presentation:** 3
**Contribution:** 1
**Rating:** 4
**Confidence:** 3

**Summary:**

This paper introduces MemOrb, a lightweight, plug-and-play verbal reinforcement memory layer designed to address the problem of LLM-based customer service agents forgetting information across sessions and repeating errors. MemOrb enables continual self-improvement without requiring costly fine-tuning by distilling multi-turn interactions into compact strategy reflections called "Orbs". These Orbs are stored in a shared memory bank (using SQLite and ChromaDB) and are retrieved at inference time via a specialized retrieval and rewriting pipeline to guide the agent's decision-making, facilitating cross-user knowledge transfer.

**Strengths:**

LLM-based agents deployed in customer service often forget information across sessions, repeat errors, and lack mechanisms for continual self-improvement. To address these limitations, the paper proposes MemOrb, a lightweight and plug-and-play verbal reinforcement memory layer. This system distills multi-turn interactions into compact strategy reflections, which are stored in a shared memory bank. These reflections are then retrieved to guide decision-making, all without requiring any fine-tuning. Experiments demonstrate that MemOrb significantly improves both success rate and stability, achieving up to a 63 percentage-point gain in multi-turn success rate.

**Weaknesses:**

1. The experiments were conducted on only one benchmark (ECom-Bench), and the number of baselines is too limited. There is no quantitative comparison against the advanced memory mechanisms mentioned in Table 1.

2. The ablation studies did not sufficiently discuss or evaluate the MemOrb's additional "Rewrite" and "Self-Reflection" modules.

3. Compared to the baseline agent, how much does MemOrb increase the total computational cost when accounting for the additional Rewrite and Self-Reflection modules?

**Questions:**

1. The paper defines an Orb as a 6-tuple that includes "emotion" , and this feature is included in the embedded document for retrieval. However, its actual impact on retrieval quality or final task success rate was not evaluated. What is the effect of this feature?

2. How sensitive is the system's performance to the hyperparameter $k$? If $k$ is increased (e.g., to $k=10$ or $k=20$), does this lead to context window overload or introduce too many contradictory reflections, thereby interfering with the Actor's decision-making and degrading performance?

---

### Note · Authors · 2026-01-16

I have read and agree with the venue's withdrawal policy on behalf of myself and my co-authors.